# A previously undescribed archaeal virus suppresses host immunity

Israela Turgeman-Grott[1,2] ✉, Noam Golan[1], Uri Neri[1], Doron Naki[1], Neta Altman-Price[1,2], Kim Eizenshtein[1], Deepak K Choudhary[1], Rachel Levy [ID][1], Sharon Navok[1], Lee Cohen[1], Yarden Shalev[1], Himani Singla [ID][1], Leah Reshef[1] & Uri Gophna [ID][1] ✉

## Abstract

**Extremophilic archaea can have chronic viral infections that are well-tolerated by the hosts and may potentially protect against more lethal infections. Here we show that a natural Haloferax strain (48N), is chronically infected by a lemon-shaped virus. This viral infection is not cleared spontaneously, despite the multiple defense systems of the host. Curing 48N of its virus led to radical changes in the gene expression profile of 48N and a dramatic improvement in its growth rate. Remarkably, the cured 48N is the fastest-growing haloarchaeon reported to date, with a generation time of ~107 min at 45 °C, and faster than any known Haloferax species at this temperature. The virus subverts host defenses by reducing its transcription, including the CRISPR spacer acquisition machinery. Nonetheless, even in the virus-cured background, spacer acquisition is very low, indicating that another genetic element is disrupting CRISPR activity. Our results suggest that the slow growth of some halophilic archaea could be due to the effects of proviruses within their genomes that consume resources and alter the gene expression of their hosts.**

**Keywords** Spindle-Shaped Virus; Phage; Chronic Infection; Haloarchaea; CRISPR-Cas
**Subject Categories** Immunology; Microbiology, Virology & Host Pathogen Interaction

## Introduction

A major challenge in microbiology is to understand virus–host interactions that involve chronic infections, where cells maintain their viability yet release virions over extended periods without clearing the infection. Viral infections are expected to lead to a major loss of fitness for the infected strains, and consequently to their demise, yet in some environments such infections can be paradoxically quite common (Munson-McGee et al, 2018; Wirth and Young, 2020). Archaeal viruses represent a true frontier for virology research, showing an enormous diversity of morphology, egress mechanisms (Baquero et al, 2021), and molecular mechanisms. Ecologically, the impact of archaeal viruses on their hosts, in some ecosystems such as deep marine sediments, may be greater than that of bacteriophages (Danovaro et al, 2016), yet relatively few of them have been cultivated (Dellas et al, 2014; Prangishvili, 2013; Prangishvili et al, 2017; Wirth and Young, 2020), compared to bacteriophages or eukaryotic viruses. Chronic infections appear to be common in archaea, and the viruses that cause such infections are often challenging to isolate since they do not form plaques on soft agar plates. Past studies of such viruses have thus relied on metagenomics and single-cell genomics to identify and quantify host-virus interactions in archaea-dominated ecosystems. These approaches have led to major discoveries, including the observation that in some extreme environments >60% of cells are infected with viruses and most of them contain genomes of two or more virus types (Munson-McGee et al, 2018). This has led to the hypothesis that in some natural environments, lysogeny (where a virus is completely dormant) or a relatively mild chronic infection is not just tolerable, but may actually be beneficial when it offers protection to cells against more virulent viruses (Munson-McGee et al, 2018; Wirth and Young, 2020). Moreover, it has also been shown in thermo-acidophilic archaea that chronic infection can be maintained when infected cells produce a toxin that kills off competing non-infected strains (DeWerff et al, 2020).

In haloarchaea, chronic viral infections are common (Alarcón-Schumacher et al, 2022; Pietilä et al, 2009; Pietilä et al, 2016), yet unlike lytic infection (Schwarzer et al, 2023), there was no good model system to study chronic infection until very recently. The newly established system of *Haloferax volcanii* pleomorphic virus 1 (Alarcón-Schumacher et al, 2022), uses a virus isolated from an environmental sample, whose original host remains unknown, to successfully infect the lab strain of *H. volcanii* (DS2), the best studied archaeal model organism (van Wolferen et al, 2022). This virus has a relatively minor effect on host growth, in line with the "tolerable infection" scenario.

Many archaea harbor multiple defense systems, including CRISPR-Cas systems, which provide acquired immunity. Chronically infecting viruses must thus avoid triggering such defenses, or subvert them, in order to establish a long-term coexistence with their hosts. In the case of *H. volcanii* pleomorphic virus 1, lack of

[1]The Shmunis School of Biomedicine and Cancer Research, Faculty of Life Sciences, Tel Aviv University, Tel Aviv, Israel. [2]The Avinoam Adam Department of Natural Sciences, The Open University of Israel, Raanana, Israel. ✉E-mail: isratur@openu.ac.il; urigo@tauex.tau.ac.il

CRISPR-mediated acquired immunity by DS2 during exponential growth was attributed to a twofold downregulation of the acquisition module gene *cas1*, yet no downregulation was observed during the stationary phase, despite sustained virus production (Alarcón-Schumacher et al, 2022).

Here, we study the interaction of a chronically infecting lemon-shaped virus and its host, the halophilic archaeon *Haloferax volcanii* 48N (henceforth 48N), which is closely related to the model organism *H. volcanii* (97.91% average nucleotide identity in coding genes). By using a "gene therapy" approach in which the provirus was deleted from the genome, we were able to cure 48N of its virus, and compare phenotypes and gene expression of the cured and infected strains. We observed that under chronic infection, growth is dramatically delayed and antiviral defense systems are transcriptionally repressed, and the CRISPR-Cas-based acquired immunity is disrupted.

## Results and discussion

### Haloferax volcanii DS2 acquires CRISPR spacers against a region within the 48N genome that encodes a provirus

48N is a strain originally isolated from the tidal pools of Atlit for which a draft genome is available (Shalev et al, 2018; Turgeman-Grott et al, 2019). We were interested to perform a mating experiment between 48N and *Haloferax volcanii* DS2 to test whether the latter can acquire genes or spacers from the former. We therefore performed a mating experiment between a uracil auxotroph derivative of 48N (UG613, Methods) that carries the pWL102 plasmid conferring resistance to mevinolin and a tryptophan auxotroph derivative of *Haloferax volcanii* DS2 (UG469). This assay, designed to obtain mostly colonies of UG469 that received pWL102 by mating, was conducted by applying stationary cultures of the two strains onto a filter on a rich medium and then transferring them to selectable plates containing mevinolin and lacking uracil. We then tested the spacer acquisition profile of the CRISPR-Cas system in the UG469 mating products and observed that spacer acquisition was highly skewed: the vast majority of spacers matched just one region in the 48N genome (Fig. 1A; Appendix Table S1). A closer examination of that genome region showed that it contained ORFs encoding two homologs of bacteriophage-associated integrases that are typical to temperate phages. The same region also had matches in the CRISPR arrays of two *Haloferax* strains isolated in the same year from the same site as 48N: *Haloferax* strains 24N and 47N (Table 1), which did not contain the putative virus locus in their genomes. Most notably, in both of these strains, the leader-proximal spacer, which in type I systems is the one most recently acquired, matched the suspected provirus (Table 1). The pattern of acquisition resembled that previously observed in bacteria following infection with dsDNA phages (Modell et al, 2017), i.e., a strong enrichment for a specific part of the viral genome, potentially representing the free end of the injected DNA (Fig. 1A). Taken together these observations led us to hypothesize that spacers were acquired from a virus that chronically infected 48N and was able to deliver its DNA into strain UG469, and probably some environmental strains of *Haloferax*, but failed to establish an infection in those strains.

To test whether the suspected genomic region indeed originated from a replicating mobile element, we first obtained a fully assembled genome of 48N main chromosome (for which only a draft was available) along with its four natural plasmids, by combining the previous Illumina reads with newly generated Pacific BioScience SMRT sequencing data (see "Methods"). We then mapped the raw Illumina reads to the genome and obtained coverage scores of different areas of the genome. This analysis showed that coverage of the suspected provirus locus was over 20-fold higher than the genome average (Fig. 1B). Notably, examination of the assembly graph showed that the provirus element exists both as an integrated provirus and in a circular form.

In addition to the CRISPR-targeted proviral region, three other genomic regions within the 48N main chromosome had sequence coverage of about 2.5–3.5-fold higher than the mean coverage, indicating that additional replicating elements are present in the 48N culture (Fig. 1B). One of these elements showed a high BLASTX sequence similarity to an IS-6 like transposase and a type iv pilin gene, indicating a putative transposable element. The second encoded an ORF with 97% BLASTX identity to the putative rep protein of Haloferax virus Halfgib1 (Dyall-Smith et al, 2021), an integrase. and an ORF related ORF1 (a viral spike protein) of Halorubrum pleomorphic viruses 2 and 6 (43% amino acid identity), and is likely to be a viral element related to pleoplipoviruses. The third of these high-coverage regions encoded ORFs that only had similarity to proteins of unknown function from haloarchaea and other euryarchaeota. Importantly, no spacers were acquired against these genetic elements by the DS2 CRISPR-Cas system, indicating that their DNA was probably not delivered from 48N to DS2 in the mating experiment.

### 48N encodes a chronically infecting lemon-shaped virus

To test whether the replicating element we identified is indeed a virus, we concentrated the supernatant of a 48N culture and submitted it to transmission electron microscopy (TEM). TEM images revealed the presence of predominantly lemon-shaped virus-like particles about 65–90 nm long and 35–50 nm wide (Fig. 2A–C), similar in morphology to lemon│ (spindle)-shaped viruses previously observed in other archaeal genera such as *Haloarcula* (Bath et al, 2006; Bath and Dyall-Smith, 1998) and *Sulfolobus/Saccharolobus* (Ceballos et al, 2012; Schleper et al, 1992; Wiedenheft et al, 2004), despite the lack of sequence similarity of any of the ORFs encoded by the provirus to any known capsid protein. Both intracellular viruses and a fraction of the secreted ones appeared attached to fibrillar structures (blue arrows, Fig. 2C,E). The supernatant also contained additional spherical virus-like particles about 75–90 nm in diameter (Fig. 2D). When we extracted DNA from the supernatant and submitted it to Illumina sequencing and de-novo genome assembly (see "Methods"), we observed: numerous small (>5kbp) contigs of relatively low coverage depth (averaging between 0.95- and 4.82-fold coverage) mapping to the host chromosome and plasmids, and two prominent contigs, 24kbp and 16kbp long, of higher coverage depth (7403.05 and 17.29 average fold, respectively (Dataset EV1), The 24kbp contig corresponded to a circular replicon. The 16kbp contig encoded two putative proteins that showed similarity to proteins conserved in haloarchaeal pleomorphic viruses (one matching ORF7 of *Halogeometricum* sp. pleomorphic virus 1

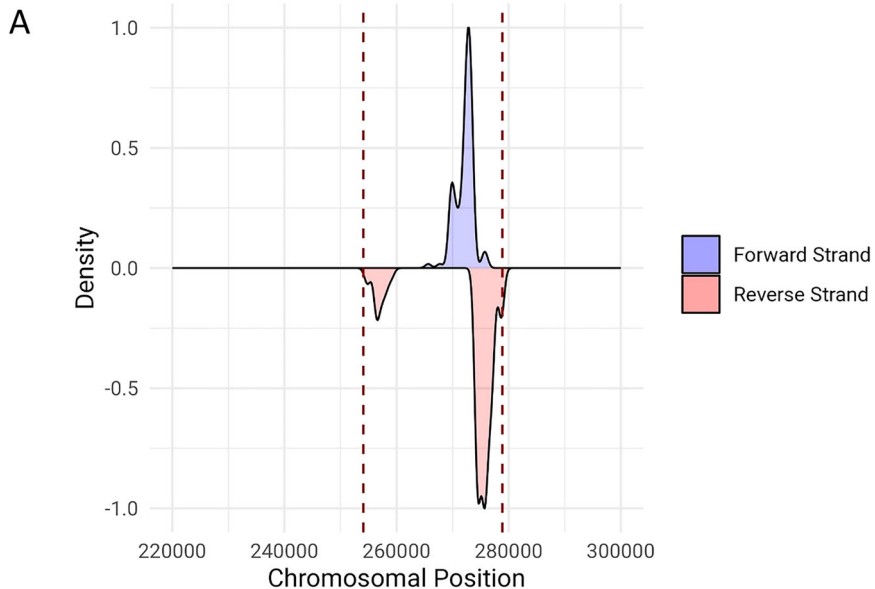

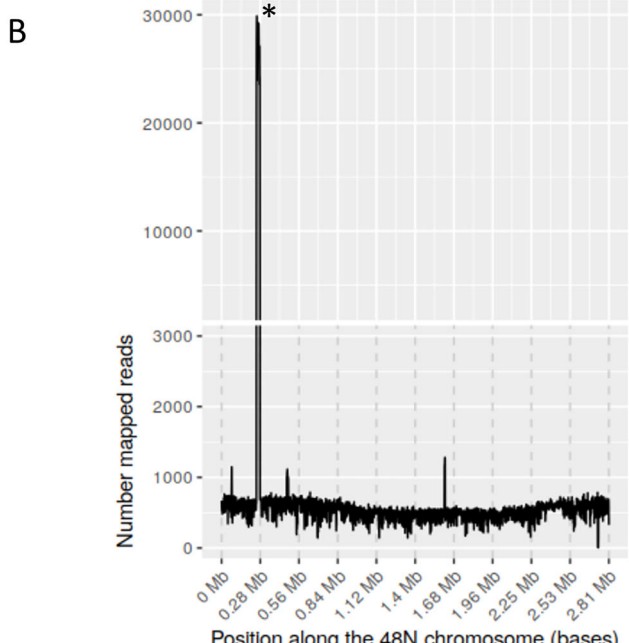

**Figure 1. Genomics-based evidence for the existence of a virus in 48N.**

(A) CRISPR spacer acquisition against the 48N chromosome by *H. volcanii* during mating on filters is almost exclusively against one locus in that genome. (B) Coverage plot of Illumina sequence reads that map to the 48N chromosome. DNA was extracted from cells grown to high cell density over 3 days (stationary culture). Asterisk marks the highest coverage peak, which corresponds to the locus from which the majority of spacers are derived.

(Atanasova et al, 2012; Sencilo et al, 2012), and the other ORF6 of that virus with a closer match to *Haloferax* pleomorphic virus 1 (Alarcón-Schumacher et al, 2022). In addition, the 16kbp contig encoded a tyrosine recombinase/integrase typical to archaeal proviruses and temperate bacteriophages that have double-stranded DNA genomes. This implies that the 16kbp contig likely belongs to a pleomorphic virus, while the 24kbp genome corresponds to a lemon-shaped virus (henceforth LSV-48N). LSV-48N virions were also observed within 48N cells (Fig. 2E).

In order to identify structural proteins of these two putative viruses, we precipitated the supernatant of 48N cells, and identified viral proteins by digestion with trypsin followed by LC-MS-MS analysis (see "Methods"). Of the proteins detected in all three biological replicates, we observed 5 that corresponded to the

**Table 1. Spacers matching the provirus locus from 48N in genomes of natural isolates obtained from the same site and season.**

| Strain | Spacer | Location in the array |
|---|---|---|
| 24N | AAAGACGTACTCTCGCTCGGCGGTCCAGTTCGGCGAC | **1/37** |
|  | ACTGAACGAGTCGCCGTTGACTTCCTGAGCGACCTGCC | 11/43 |
| 47N | GTTGATGACGACAGCGCAGGACTTGGACTCGACTTCAG | **1/34** |
|  | TCACTGCTCATCACCCAACTCAAGAGCCATGACTAACG | 2/34 |
|  | ATGGTCAACCTCCTCTGCAACGGGCTCATGACCGC | 3/34 |

Leader-proximal spacer locations are highlighted in bold.

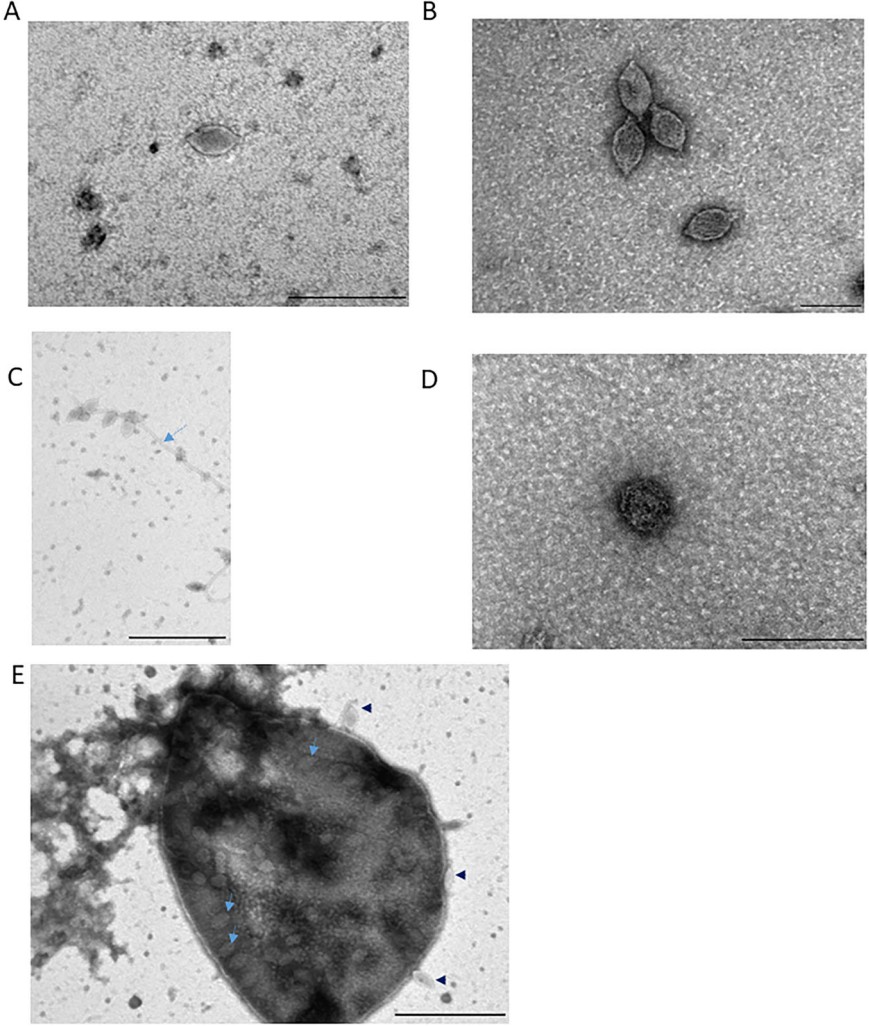

**Figure 2. Viruses infecting Haloferax Atlit 48N.**

(A–C) Transmission electron micrographs of lemon-shaped viruses released to the supernatant. Length bars correspond to 200, 100, and 500 nm, respectively. (D) A pleomorphic virus present in the same supernatant preparation. Length bar corresponds to 200 nm. (E) Lemon-shaped viruses outside (black triangles) and within cells. Blue arrows mark fibrillar structures that some of the viruses adhered to. Scale bars correspond to 500 nm. Source data are available online for this figure.

putative pleomorphic virus (the 16Kb contig) and 13 that matched LSV-48N (Dataset EV2). Using intensity-based absolute quantification (iBAQ), we inferred that the most abundant protein in the supernatant encoded by the putative pleomorphic virus was a homolog of the envelope protein of Halorubrum pleomorphic virus 1 (Pietilä et al, 2009) (amino acid identity and similarity of 46% and 62%, respectively) with iBAQ of $5.3 \times 10^7$, while the most abundant LSV-48N encoded protein was a protein of unknown function iBAQ of $4.61 \times 10^8$. HHPRED analysis (Soding et al, 2005; Zimmermann et al, 2018) showed the latter protein to have a

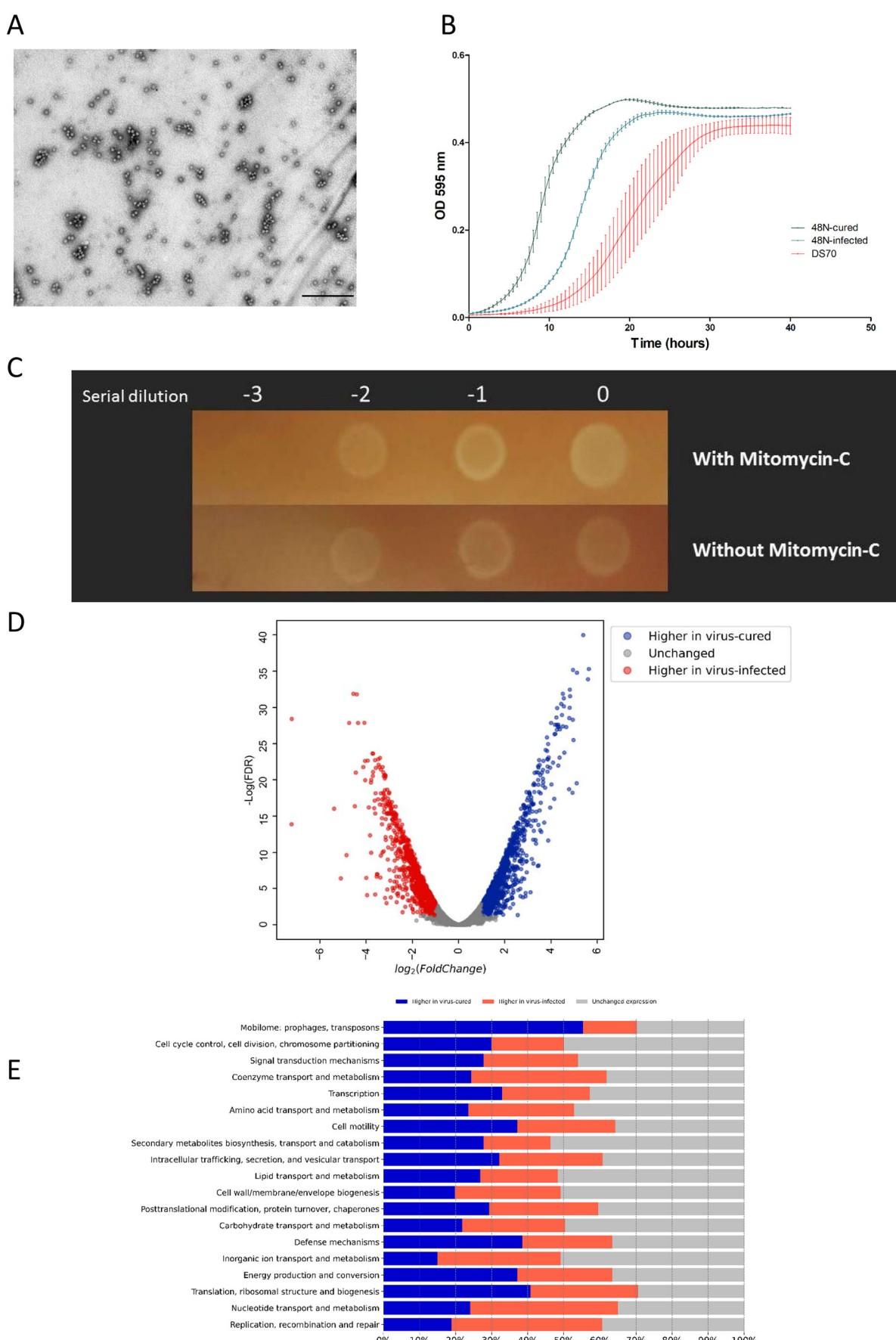

**Figure 3.   Effects of virus curing on gene expression and growth of the host.**

(A) The supernatant from cured-48N cells contains spherical virus-like particles. (B) growth curves of 48N strains and the *H. volcanii* lab strain. Values represent means of three biological replicates and error bars the standard error of the mean. Samples were grown overnight in YPC medium at 45 °C to the mid-log phase and then diluted with a fresh medium to $OD_{595\,nm}$ of 0.1. The growth curves were carried out in microplate reader at 45 °C with continuous shaking while measurement were taken every 30 min at a wavelength of 595 nm. (C) Spot assay measuring plating efficiency following mitomycin C induction. The image shown represents a single comparative experiment. (D) A volcano plot showing differentially abundant transcripts in cultures of, 48N and virus-cured 48N based on RNA-Seq, each the mean of three biological replicates. (E) Gene categories affected by infection with LSV-48N. Source data are available online for this figure.

structural similarity in its C-terminal end to a tail protein of a Sulfolobus spindle-shaped virus 19 (SSV19 (Han et al, 2022), probability 92.36%, but with a poor *E*-value of 4.9), while residues 365–549 had structural similarity to a bacterial beta-sandwich, cellulosome protein [(Brunecky et al, 2012), probability 98.09%, *E*-value: 0.002]. Another protein of unknown function present in the supernatant in lower amounts also had HHPRED-based structural similarity to a structural protein of SSV19 (70.2% probability, *E*-value: 56).

These results confirm that both viruses are released from cells, and that LSV-48N is probably more abundant than the pleomorphic virus. Importantly, we did not detect proteins from the putative 12 kb element, in agreement with the lack of detection of its genome in the supernatant.

## LSV-48N is non-lethal and is continuously shed during growth

We used outward-facing primers that can amplify a circularized genome or primers that amplify the edges of the integrated form of the virus (see "Methods") for PCR-based analysis of DNA extracted from 48N cultures. This analysis indicated that both circular and integrated forms are present under standard, lower-salt and low-phosphate conditions (Appendix Fig. S1), previously shown to lower the chromosome copy number in Haloferax species that are highly polyploid (Stachler et al, 2017; Zerulla et al, 2014). In order to determine at what stages during 48N growth curve are viruses produced and secreted, we used droplet digital PCR (ddPCR) to quantify viral genome copies in DNA extracted from the supernatant as well as from the cells at different time points when growing 48N in rich liquid medium (see "Methods"). Results showed that during early exponential growth (OD of 0.1) there is maximal accumulation of viral genome copies within cells, yet the maximal release of viruses to the medium occurs during late exponential growth-early stationary phase (OD of 0.7–1) (Fig. EV1). When the viability of the cells was tested using fluorescent microscopy and the LIVE/DEAD™ BacLight™ kit, there were very few dead cells present (roughly 1.5%, Appendix Fig. S2). Furthermore, 48N consistently grew faster than *H. volcanii* lab strain (Fig. 3B), indicating that chronic infection with LSV-48N is well-tolerated by the host.

## LSV-48N is a representative of a new family of viral elements in Haloferax

Since LSV-48N showed no significant sequence similarity to known haloarchaeal viruses, we used sequence comparisons of six-frame translations of LSV-48N sequences to compare with available haloarchaeal genomes to identify related integrated proviruses. We detected several related proviruses in other strains and species of *Haloferax* that had notable similarities in gene organization and sequence to LSV-48N (Fig. 4). Given that nearly all LSV-48N genes had homologs in these proviruses, we conclude that LSV-48N is the first studied representative of a new family of viruses able to integrate into the genomes of *Haloferax* species.

## LSV-48N profoundly affects the growth and physiology of its host

Even when grown for 400 generations in rich medium, no colonies that spontaneously lost LSV-48N (tested by virus-specific colony PCR) could be detected. We therefore attempted to delete the provirus from the genome by deleting the putative phage-like integrase from UG613 first, using the pop-in/pop-out approach (Bitan-Banin et al, 2003). However, after obtaining a few colonies on a medium lacking uracil (see "Methods") and then switching them to a low-phosphate medium containing uracil and 5FOA, long-read genome sequencing indicated that LSV-48N was deleted entirely from the genome, and the replicative form disappeared. TEM images showed lemon-shaped viruses in the supernatant of the infected 48N strain but not in the virus-cured strain (cured 48N), while the spherical virus-like particles were still present (Fig. 3A).

Chronic viral infection often results in a considerable growth burden and a major shift of the host gene expression (Alarcón-Schumacher et al, 2022). Indeed, when we compared the growth of the wild-type-infected 48N to its cured 48N, we observed a major growth improvement, with a much improved growth rate, the fastest to our knowledge ever observed for a Haloferax species (Robinson et al, 2005) at 45 °C (Fig. 3B; Movie EV1). Chronic viruses that delay the growth of the hosts often produce turbid plaques. We attempted to induce higher virus load by mitomycin C induction and tested this using a newly developed highly sensitive plaque assay developed specifically for chronic haloarchaeal viruses (Navok et al, 2025). In this assay, we used the cured strain as host to indicate successful infection by plaque formation. Indeed, a slightly higher plating (plaquing) efficiency was observed after induction when using the same volume of supernatant from mitomycin C-induced culture (Fig. 3C).

We then extracted RNA from both naturally-infected and cured strains in the logarithmic growth phase and compared their transcriptomes using RNA-seq (Dataset EV3). As can be expected from such striking differences in growth, RNA-seq analysis revealed an immense difference in the transcription pattern of 48N, with almost 60% of the genes in cured 48N exhibiting a significant difference in their expression (Fig. 3D) that spanned all functional categories (Fig. 3E). The large increase in generation time in the virus-infected strain indicates that the virus is

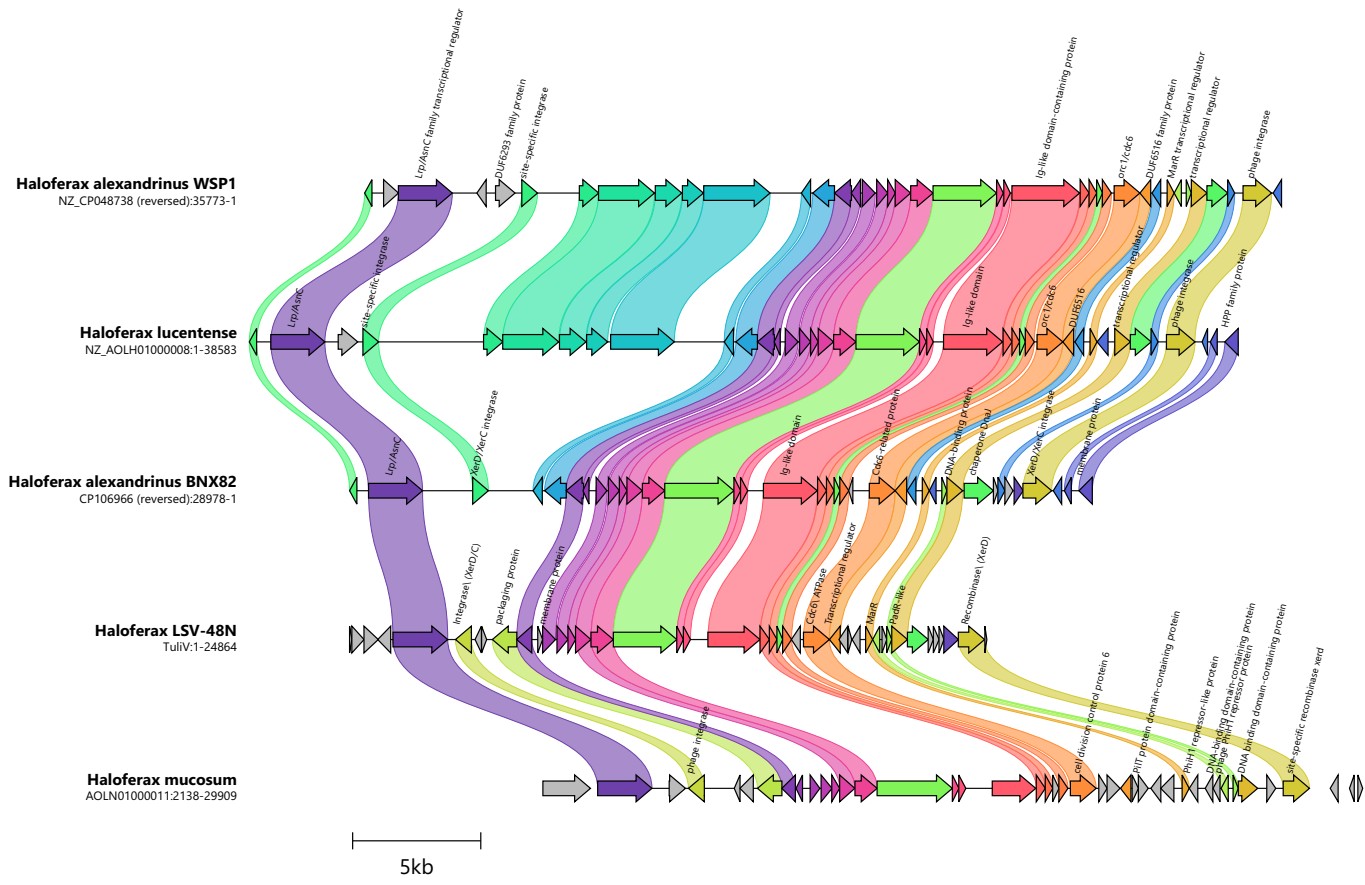

**Figure 4. Comparative synteny plots of putative *Haloferax* proviruses.**

Gene cluster plot was generated with clinker (Gilchrist and Chooi, 2021). Each color represents a gene family, and linked protein-coding genes have over 50% pair-wise sequence identity.

consuming much of the cellular resources and potentially modulating host metabolism. The RNA-seq results showed that not only were viral genes encoding membrane proteins very highly transcribed (Fig. EV2), but the expression of many host genes which are associated with different functions, including metabolism, were unexpectedly altered. Thus, the two cytochrome bd-type quinol oxidase subunits that play a critical role in respiration in haloarchaea were more than tenfold higher in the infected strain, implying that respiration may be increased during infection. In contrast, many host metabolic genes were strongly downregulated, such as the archaeal-type H+ ATPase subunits, nearly all of which had mRNA levels decrease of more than 15-fold.

## LSV downregulates the defense systems of its host

Based on genomic analysis using the PADLOC server (Payne et al, 2022), the 48N genome contains multiple potential antiviral defenses encoded on its natural plasmids. Among those, CRISPR-Cas and the two CBASS systems should be able to prevent chronic viral infection: CRISPR-Cas can generally acquire spacers against replicating selfish elements and then target their DNA for degradation, while CBASS systems cause abortive infection by recognizing viral infection and killing the host (Millman et al,

2020). However, a comparison of the expression levels of defense genes in 48N and cured 48N showed that these defense systems, and additional ones, such as Hachiman, were strongly down-regulated in the infected strain (Table 2). Notably, the genes encoding the cyclases of the CBASS systems, required for their activation in the presence of viral infection had much lower levels in 48N compared to cured 48N (8.29- and 14.6-fold lower in CBASS type 1 and 2, respectively), as were the *cas2* and *cas4* genes required for new spacer acquisition (17.19 and 4.68 lower, respectively). Thus, LSV-48N infection downregulates cellular defenses, either directly or indirectly at the level of transcription, sufficiently to allow chronic and sustained infection.

To further investigate why 48N does not acquire CRISPR immunity against LSV-48N, we performed a spacer acquisition assay on the 48N CRISPR arrays using amplicon sequencing, comparing the infected and virus-cured strains. For both strains, we observed extremely low spacer acquisition levels, reflected by a low ratio of newly acquired spacers to ones that were mere duplications of previously existing spacers in the respective arrays (between 5% and 0.06%, Appendix Table S2). This ratio was markedly higher in the cured strain for arrays A and B, but not for array C (Fig. EV3). Up to 6% of the duplicated spacers had multiple mutations, indicating an unknown underlying process that results in spacer

**Table 2. Antiviral immunity genes whose expression is significantly higher in the cured strain vs. its infected isogen.**

| Predicted protein encoded by the gene | Level in cured (RPKM) | Level in Infected (RPKM) | Fold increase (cured) |
|---|---|---|---|
| CRISPR-associated protein Cas7 | 205.31 | 53.85 | 3.81 |
| CRISPR-associated protein Cas5 | 146.69 | 25.36 | 5.79 |
| CRISPR-associated protein Cas4 | 95.30 | 20.38 | 4.68 |
| CRISPR-associated protein Cas2 | 126.33 | 7.35 | 17.19 |
| CBASS_1_Cyclase | 45.85 | 5.53 | 8.29 |
| CBASS_1_Effector | 66.68 | 23.89 | 2.79 |
| CBASS_2 Cyclase | 343.63 | 23.56 | 14.58 |
| DUF1837 domain-containing protein/Hachiman A1 | 347.90 | 13.25 | 26.25 |
| DEAD/DEAH box helicase/ Hachiman B1 | 222.52 | 30.53 | 7.29 |

sequences that are sometimes no longer detectable by standard BLASTN thresholds (Appendix Fig. S3). In terms of genomic location from which spacers were derived, the pattern was fairly similar between virus-cured and w.t., and no spacers that originated from the genome of LSV-48N (Fig. EV4). For the spacers that were newly acquired from the 48N genome, a clear TTC PAM signature could be detected, similar to that observed previously for *H. mediterranei* or *Haloarcula hispanica* (Fig. EV5). Thus, while LSV-48N somehow reduces the transcription of *cas* genes, it is probably another genetic element within the 48N genome that more strongly interferes with the spacer acquisition process of the 48N CRISPR-Cas system.

## Conclusions

The lemon (spindle) viral morphotype is unique to the domain archaea and is the most abundant particle shape in several hypersaline ecosystems (Oren et al, 1997; Sime-Ngando et al, 2011). Surprisingly, LSV-48N is unrelated by sequence (some structural similarity in predicted proteins does exist, see above) to previously isolated haloarchaeal lemon-shaped viruses (Bath et al, 2006; Bath and Dyall-Smith, 1998), but rather shares several characteristics with fuselloviruses of Thermoproteota (such as SSV1 and SSV2), despite lack of significant sequence similarity, having a circular double-stranded genome, causing no cell lysis, and existing both as circular plasmid-like form and as an integrated provirus (Aulitto et al, 2022; Schleper et al, 1992). Furthermore, infection by SSV1 was shown to dampen the induction of CRISPR-Cas systems in *Saccharolobus solfataricus*, similar to what we show for LSV-48N (Fusco et al, 2015). However, unlike SSV1 that has a "harmonic coexistence" with its host, with only minor effects on host gene expression (unless induced by UV), and little effect on growth (Aulitto et al, 2022; Schleper et al, 1992), LSV-48N causes drastic changes in both host gene expression and growth. It should be noted that the unrelated virus HFPV-1, while having only a modest effect on growth also affected the expression of over a thousand

genes, indicating that chronic viral infection will profoundly affect the archaeal host, even when the growth burden is minimal.

LSV-48N encodes four putative transcriptional regulators that have strong and significant HHPRED structural similarity to known regulators: a helix-turn-helix protein with similarity to the MecI repressor of *Staphylococcus aureus* (PDB 1OKR, E-value 8.4e-7, 98.53%), a PadR-like transcriptional regulator similar to the *Bacillus cereus* repressor BC4206 (PDB 4ESB E-value 4.4e-10, probability 99.18%) and a viral winged helix protein from Sulfolobus Turreted Icosahedral Virus (PDB 2COS_A, E-value 2.1e-10, probability 99.22%), a helix-turn-helix protein DNA binding protein structurally similar to the important transcriptional regulator RosR from the haloarchaeon *Halobacterium salinarum* (PDB 6F5C_B, E-value 9.9e-10, probability 99.06%) (Kutnowski et al, 2018; Tonner et al, 2015), and a swapped hairpin fold protein similar to the ARB-like transcriptional regulator from the hybrid virus-plasmid pSSVx a satellite virus of the Sufolobales spindle-shaped virus SSV2 [(Contursi et al, 2011). PDB 3O27_A, E-value 0.031, probability 96.38%]. It is possible that one or more of these regulators can bind host DNA sequences and not just viral ones, thereby causing some of the observed transcriptional effects of curing the host of LSV-48N. The ability to downregulate transcription of antiviral immunity genes is common in mammalian viruses (Katze et al, 2008), including by Ebola and Marburgviruses (Kash et al, 2006), Rabies virus (Wang et al, 2005), Pseudorabies virus (Brukman and Enquist, 2006), and entero/rhinoviruses (Tsalik et al, 2021). In contrast, bacteriophages are known to use less specific suppression mechanisms, such as strong downregulation of all genes transcribed by the major bacterial sigma factor σ70 (Hinton et al, 2005; Liu et al, 2014), shutting down transcription from alternative sigma factors (Brown et al, 2016; Tabib-Salazar et al, 2018), or general destabilization of host mRNAs (Ueno and Yonesaki, 2004). LSV-48N in this respect may be somewhere in between mammalian viruses and bacteriophages in terms of its effects on host transcription. Additionally, since virus curing resulted from an unclear molecular process (see "Methods"), it is possible that some changes to gene expression are the product of secondary heterozygous mutations that due to the highly polyploid nature of this organism, escaped detection by sequencing.

Extreme environments that are often dominated by archaea tend to have lower diversity and lower cell density than bacteria-dominated hyperdiverse ecosystems such as soil or the mammalian gut. Consequently, host-virus interactions of extremophilic archaea may differ greatly from phage-bacteria interactions. Curing bacteria of multiple lysogenic phages often produces substantial physiological changes, far beyond sensitivity to viral infection. In some bacteria such as *Lactococcus lactis* pro-phage curing resulted in improved growth and antimicrobial resistance (Aucouturier et al, 2018), while in *Escherichia coli*, the opposite was observed (Wang et al, 2010), and in *Staphylococcus aureus* deletion of multiple prophages led to inability to cause disease in a mouse model (Bae et al, 2006). In other cases, such as in *Streptococcus pyogenes*, no differences in growth rate were observed (Euler et al, 2016). However, LSV-48N is a more complex case, since it exists both in replicating and integrated forms, probably because Haloferax species are highly polyploid. It has been suggested that in extreme environments, when chronic infection protects against lethal infection, it can actually be beneficial. The viruses in such

environments, in turn, may benefit from a chronic lifestyle when the host cell density is low, making long-term exploitation of the host and its progeny much more desirable (Weitz et al, 2019). Here, we examine the physiology and gene expression of a chronically infected extremophile strain. Importantly, because it was isolated directly from an evaporation pond where other strains of the same species abound, it can be assumed to be relatively fit, despite being infected. Furthermore, it even grows faster than DS2 in the lab on standard media (Fig. 3B). Nonetheless, the seemingly mild burden of chronic infection is in reality not mild at all—curing the virus dramatically increased the growth rate as well as the expression of antiviral defense genes. While it may seem paradoxical that such a burden is sustained in a natural habitat, one should keep in mind that in many ecosystems that are continuously limited for resources, such as available carbon and nitrogen, and are not subject to feast or famine fluctuations, the maximal growth rate may not be the key trait determining ecological fitness. Furthermore, the two strains that were isolated in the same sampling round with 48N (24N and 47N) that were not infected by LSV-48N and had CRISPR spacers matching its genome, had similar growth rates to that of 48N (generation times on rich medium of 2.08 and 2.96 h. For 24N and 47N, respectively, vs. 2.18 h for 48N). Thus, as observed previously for other extreme environments, hypersaline systems may allow chronic infection with viruses to be tolerated (Munson-McGee et al, 2018; Wirth and Young, 2020).

## Methods

### Reagents and tools table

| Reagent/resource | Reference or source | Identifier or catalog number |
|---|---|---|
| **Haloferax strains** | | |
| Haloferax strain Atlit 48N | Shalev et al, 2018 | N/A |
| *Haloferax volcanii* DS2 | Hartman et al, 2010 | |
| *Haloferax volcanii* UG613 | This study | |
| *Haloferax* strain UG685 (48N virus-cured) | This study | |
| *Haloferax volcanii* UG469 (*ΔtrpA, HVO_0894::pyrE*) | This study | |
| **Oligonucleotides** | | |
| Oligonucleotides | This study | Appendix Table S3 |
| **Plasmids** | This study | Appendix Table S4 |
| **Software** | | |
| PEAR | Zhang et al, 2014 | N/A |
| bbtools | https://sourceforge.net/projects/bbmap/ | N/A |
| bowtie2 | Langmead and Salzberg, 2012 | N/A |
| Trimmomatic | Bolger et al, 2014 | N/A |
| SPAdes (version 3.11.0) | Prjibelski et al, 2020 | N/A |
| Unicycler | Wick et al, 2017 | N/A |
| CheckM | Parks et al, 2015 | N/A |
| HHpred | Zimmermann et al, 2018 | N/A |
| CollabFold2 | Mirdita et al, 2022 | N/A |
| Genemark | Zhu et al, 2010 | N/A |
| hmmsearch | McClure et al, 1996; Wistrand and Sonnhammer, 2005 | N/A |
| mmseqs | Steinegger and Söding, 2017 | N/A |

## Culture conditions

Haloferax strains were routinely grown aerobically at 45 °C in either Hv-YPC (rich medium) or in Hv-Ca/Hv-Ca+ (minimal medium) as described in (Allers et al, 2010). When required, uracil was added at a concentration of 50 μg/ml. For the pop-out selection medium, 5-fluoroorotic acid (5FOA) was added to a concentration of 50 μg/ml.

## Droplet digital PCR (ddPCR) analysis on circular LSV-48N

The number of viral particles in 1 ml of 48N culture was inferred from ddPCR experiments (Bio-Rad ddPCR device according to manufacturer's protocol). In all, 1 ml from cultures at different growth stages ($OD_{600\,nm}$: 0.1, 0.2, 0.3, 0.5, 0.7, 1.0) was analyzed. In order to measure the amount of secreted viral particles in the media, DNA was extracted from 200 μl of the supernatant using Quick-DNA_viral_kit (Zymo Research) following centrifugation at 12,000 rpm. For "in cells" viral particles quantification, 1 ml from the cell culture from each growth stage was vortexed and washed twice with Hv-Ca broth to remove cell-adhered viruses. The cells were then diluted and lysed (suspended pellet cells from 1 ml with 100 μl ST buffer [1 M NaCl, 20 mM Tris.HCl] and 100 μl lysis buffer [100 mM EDTA pH 8.0, 0.2% SDS]) before viral DNA was extracted from lysed cells (Quick-DNA viral kit). Viral DNA extractions were diluted to a final concentration of 0.2 pg/μl before added to the ddPCR reaction mix. Primers used are: IT24, IT25 for the circular form of the virus and: Is660 Is661 for control housekeeping gene. 48N cells were grown to $OD_{600\,nm}$ of: 0.1, 0.2, 0.3, 0.5, 0.7, 1.0. in YPC. CFU counts were made for each culture in order to deduce the average number of archaeal cells in 1 ml.

## LSV-48N particles imaging

LSV-48N particles were visualized using a negative staining approach: Samples were adsorbed on formvar/carbon-coated grids and stained with 2% aqueous uranyl acetate. Samples were examined using JEM 1400plus transmission electron microscope (Jeol, Japan).

## Growth curves

To compare the growth of the 48N strains, each sample was grown overnight in appropriate media at 45 °C to the mid-log phase and then diluted to a fresh medium to $OD_{600\,nm}$ of 0.1. The growth curves were carried out in 96-well plates at 45 °C with continuous

shaking, using the Biotek ELX808IU-PC microplate reader. Optical density was measured every 30 min at a wavelength of 595 nm.

## LSV-48N circular/integrated form detection by PCR

Integrated and circulated form of LSV-48N inferred using PCR analysis with specific PCR oligonucleotides. The integrated form was inferred when a PCR product of 1000 bp was visible after gel electrophoresis using IT556 and IT557 for the upstream integration site and IT558 and IT559 for the downstream integration site. One primer corresponds to the genome and the other one to the virus in the integration site. The circular form was obtained using IT560 and IT561. The primers matched the ends of the virus, and the 1000 bp product is only obtained if the virus closes in a circle. PCR was performed using KAPA 2G ready mix by Roche.

## Live/dead staining

To assess the viability of 48N and virus-cured 48N a Live/Dead staining experiment was performed utilizing the LIVE/DEAD BacLight Bacterial Viability Kit (Thermo-Fisher). Mixed Cultures of 48N and 48N cured were grown overnight in YPC medium to late logarithmic growth in multiple biological replicates. Subsequently, the cultures were diluted to an optical density (OD) of 0.1 and ~50 μl of resuspended cultures were aliquoted into 1-ml microfuge tubes. Subsequently, 10 μl of Propidium iodide dye and 10 μl of SYTO9 dye were added separately from stock solutions. The microfuge tubes were covered with aluminum foil to prevent exposure to light and minimize dye degradation. The treated cultures were then incubated in the dark for 15–20 min to facilitate cellular uptake of the stains. Following the incubation period, 10 μl of the treated culture was dispensed onto a glass slide, and was visualized using a confocal microscope. Multiple images were acquired at magnifications of ×20 and ×40 to visualize the stained cells and assess their viability.

## Mating experiments

Liquid cultures of both parental strains were grown to an $OD_{600}$ of ~1.8. The parental strains were then mixed in 1:1 ratio and applied to a nitrocellulose 0.45-μm filters using a Swinnex 25-mm filter holder. The filter with the mating products was transferred to a rich medium plate (Hv-YPC) for 24 h for phenotypic expression. The cells were then resuspended and washed in Hv-Ca broth before plated on Hv-Ca media containing mevinolin and lacking uracil. After 14 days in 45 °C incubator, we were able to detect colonies that grew on the selectable plates. In each biological replicate, about 200 colonies were picked, and their DNA was extracted using the spooling protocol (Allers et al, 2004). The DNA samples were used to identify newly acquired CRISPR spacers (see below).

## New CRISPR spacers detection

Extracted DNA from the 48N-DS2 mating was used as PCR template for either *H. volcanii* or 48N CRISPR arrays using specific primers amplifying the region between the leader and the third spacer of each array (Appendix Table S3). When analyzed by agarose gel electrophoresis, PCR products 70 bp higher than the main PCR fragment represent potential new spacer-repeat acquisitions. We extracted from the gel the region that corresponded to amplicons of increased length (potentially having the new spacer sequences), isolated the DNA from the gel and amplified the fragment through another cycle of PCR. New acquisition events could then be detected via the presence of a higher band in the gel. PCR products were then sent for further processing and Illumina amplicon sequencing (240,000–290,000 reads per array per biological repeat) at the Center for Genomic Research, University of Illinois, USA. Briefly, the elongated PCR product was enriched using Ampure beads size selection; sample specific barcodes and Illumina adapters were added by PCR; and the resulting products were purified, pooled, and paired-ends sequenced on a MiSeq Illumina platform. Notably, even after these consecutive steps of size selection, many reads still represented amplicons derived from no acquisition amplifications. Two *H. volcanii* arrays (C and D) were processed from five biological replicates in the vol-48N mating experiment; 3 48N arrays (A, B, C) were processed from four biological replicates from the 48N experiment (48N culture grown in YPC medium).

For the self-spacer acquisition experiments, 48N and 48N cured cells were cultured separately in a rich medium (Hv-YPC) for 4 days in 45 °C in shaking conditions. Subsequently, the cells were precipitated, and DNA was extracted using the QIAGEN DNeasy Blood and Tissue Kits for DNA Isolation. The isolated DNA was then utilized for subsequent analysis to identify novel CRISPR spacers.

## Processing of sequence data and spacer extraction

Raw reads were quality-filtered (Q > 20) and merged using PEAR (Zhang et al, 2014), then filtered to keep reads longer than 300 bp (corresponding to at least one event of new spacer acquisition/duplication) using bbtools (https://sourceforge.net/projects/bbmap/). A sample identifier was added to each read ID to facilitate downstream analysis. A set of custom-made R functions was used to count the number of repeats in each read, tag reads with higher repeat numbers than in the original array, and extract the sequence between the two repeats closest to the leader end. Extracted fragments of aberrant length (< 10 bp or >45 bp) were excluded from analysis. Duplicated spacers, the result of rearrangement of existing spacers within or between arrays, were identified by screening the extracted spacers against all original CRISPR arrays, allowing up to ten mismatches. Only spacers with no resemblance to existing CRISPR arrays were considered as new acquisitions.

## Mapping new spacers to the genomic location

In order to establish the protospacer location for each new acquisition, we used the blastn-short program against a reference database containing *H. volcanii* and 48N, including their natural plasmids and any additional plasmid present in the experiment (pWL102 in the mating experiment). Blast results were filtered in R with strict criteria (percent identity >95%, num. mismatch <4, length (alignment)/length (query) >0.9 and length difference between alignment and spacer <3). Blast matches fulfilling this criteria all had blast e-values lower than e-108 (median *E*-value = e-12). In cases of multiple matches, the location with the highest bitscore was preserved; if multiple matches had an equal score, target was excluded from coverage and PAM analysis.

## Generating a ura- 48N strain

Strain 48N was verified for its sensitivity to 5FOA (150 μg/ml) in YPC medium and its ability to grow on CA medium. Knockout of the *pyrE2* gene was attempted with pGB68 (Bitan-Banin et al, 2003) by pop-in and then pop-out as described in the reference. Pop-in transformants were selected on YPC plates containing Novobiocin (0.3 μg/ml) and individual colonies were then transferred into liquid YPC medium with no novobiocin and shaken in 45 °C for 48 h after which they were streaked on YPC plates supplemented with 5FOA. Resistant colonies were picked and streaked on CA plates and YPC plates. Notably, half of the colonies retained their ability to grow on CA while' resistant to 5FOA. To obtain true knockout colonies, three cycles of 5FOA selection were performed. The genotype of the pop-in colonies, as well as knockout colonies was confirmed by PCR using the primers:IT748, IT749 that flank the deleted region. Notably, this approach does not rule out heterozygosity, which could manifest in reversion to *ura+* phenotype.

## Obtaining the virus-cured 48N strain

In the pop-in/pop-out protocol (Bitan-Banin et al, 2003), the upstream and downstream flanking regions of the sequence to be deleted are amplified by PCR and cloned together into the 'suicide plasmid' pTA131, that cannot replicate autonomously and carries the *pyrE2* selectable genetic marker. The plasmids are then transformed using the polyethylene glycol, in this case into a *ura*- 48N. This was attempted in order to delete the viral integrase gene. At the stage of selection on casamino medium lacking uracil, we obtained colonies that, according to the protocol, should have contained the plasmid integrated into the genome at the flanking region. We then proceeded to counter-selection with 5FOA under low-phosphate conditions, and obtained a virus-cured strain, with a perfect deletion of the provirus. Later, deep sequencing using Nanopore technology revealed that the plasmid used for pop-in was completely absent from the genome, at the stage where colonies were obtained on a medium containing uracil. Thus, the molecular process that led to virus curing remains unclear.

## Preprocessing of Illumina and PacBio sequencing data

The Illumina raw reads used for the previous, publicly available, assembly of *Haloferax volcanii* spp. 48N were fetched. Reads mapping to PhiX174 (a common library preparation spike-in) were identified using bowtie2 (Langmead and Salzberg, 2012). Subsequently, unmapped reads were filtered using Trimmomatic (version 0.36) (Bolger et al, 2014) using the command 'adapters/NexteraPE-PE.fa:2:30:10 LEADING:3 TRAILING:3 SLIDINGWINDOW:4:15 MINLEN:36'. PacificBio Science ("long") reads were filtered using bbduk.sh from the bbtools suite (Bushnell, 2014) with the following flags: minlen=100 maq=8, i.e., reads were retained only if they were of a minimal length of 100 nucleotides and had a minimal average quality of 8.

## Assembly of 48N and LSV-48N genomes

LSV-48N processed reads were assembled using SPAdes (version 3.11.0). Manual examination of the assembly graph, together with PCR amplification, was used to verify the genome terminals and the circular topology.

The hybrid assembly of Haloferax volcanii spp. 48N was performed using Unicycler (version 0.4.4, Wick et al, 2017) with default options, supplying Pilon for genome polishing (Walker et al, 2014). The assembly graph was examined, and the topology of the virus form was manually resolved. Mapping of the raw reads and manual examination of the coverage suggested both integrated and circular forms of the virus were present in the original culture. For simplicity, the resolved form used for coverage (depth) calculation was set as the circular form (i.e., as a stand alone replicon as a separate contig) in order to report the virus coverage separately from that of the host chromosome.

Coverage reports (including estimated depth) where generated using bbmap.sh from the bbtools suite (Bushnell, 2014), via the "covstats" output. For these estimation, only the Illumina reads were provided as a query and the resolved 48N genome as reference.

Finally, the hybrid assembly of *Haloferax volcanii* spp. 48N completeness was estimated at 99.57% with contamination of 0.13% (CheckM version 1.0.18, (Parks et al, 2015)).

## Genome structural and functional annotations

For gene prediction, the assembled genomes were analyzed using genemark (Zhu et al, 2010). Predicted protein sequences were then searched (using hmmsearch (McClure et al, 1996; Wistrand and Sonnhammer, 2005), mmseqs (Steinegger and Söding, 2017) and psi-BLAST (Altschul et al, 1997)) against several public protein domain databases (Pfam, PDB, CDD, and arCOGs (Makarova et al, 2015)). The functional/descriptive label of profile matches with good alignment statistics was cloned onto the query protein. Predicted proteins without sufficiently reliable match were manually examined further using web-hhpred (Zimmermann et al, 2018) and CollabFold2 (Mirdita et al, 2022).

## Live-cell imaging

The experimental setup utilized a microfluidics chip, crafted from polydimethylsiloxane (SylgardTM 184 Silicone Elastomer Kit, Dowsil), along with its curing agent. This chip was cast on a SU-8 mold (designed by Klayout, F-Si) with the following specifications: Eight parallel flow channels (200 μm wide, 22 μm deep), divided into two smaller channels (100 μm wide, 22 μm deep). The latter comprised 144 equally spaced chambers, featuring varying dimensions (10, 20, 40, 60, 80, 100 μm wide, 60 μm deep, 0.83 μm height) on their lateral sides. To secure the microfluidics chip, a round cover glass (diameter = 47 mm, # 1.5; Menzel-Gläser, Braunschweig, Germany) was employed and affixed using an oxygen plasma cleaner (PDC-32G-2 Plasma Cleaner, Harrick Plasma, New York, USA). The archaea cultivated within this microfluidic system underwent observation and monitoring through a Nikon Ti inverted microscope equipped with a PFS (Perfect Focus System), an Andor clara CCD camera (Andor Tech., Belfast, UK), and an OKOlab Microscope Incubation System (OKOlabs GmbH, Italy). The microscope setup included a motorized stage, automated stage controllers (Nikon), shutters (BD, USA), and a Plan Apochromat Lambda D ×100 Oil immersion objective (Nikon), all controlled by the NIS Elements software.

The experimental protocol involved growing virus-infected 48N and virus-cured 48N, in YPC medium until they reached the stationary phase. Subsequently, each strain was individually introduced into a separate main channel and allowed to grow in YPC medium at 37 °C for a duration of 20 h. Following this incubation period, chambers containing these strains were selected, and a time lapse of 19.5 h was recorded at ×100 magnification, with a timer interval of 5 min between each frame. Upon completion of the experiment, data were collected and analyzed using Python scripts.

### RNA-Seq

Total RNA was extracted from four replicates of logarithmic cultures of virus-cured 48N, and three replicates of wt 48N, using the RNeasy kit (Qiagen). Extracted RNA was shipped in dry ice to the DNA Services facility within the Research Resources Center (RRC), University of Illinois at Chicago.

Libraries were prepared with NuGen's Universal Prokaryotic kit, including a rRNA depletion step, and sequenced (2x150bp) on a NovaSeq6000 SP. Raw sequencing data was processed using BBtools tool suite (sourceforge.net/projects/bbmap/,our full pipeline uploaded to https://github.com/UriNeri/LSV). In brief, bbduk was used for quality filtration, adapter trimming, removal of known sequencing artifacts such as phiX, and exclusion of reads mapping to the rRNA operon, which could not be entirely eliminated at the rRNA depletion step. BBtools sequence expression analyzer 'seal' was then used to map cleaned data to 48N coding sequences and calculate RPKM values. A single outlier replicate of virus-cured 48N was discarded at this stage, leaving 3 wt vs. 3 virus-cured replicates for differential gene expression analysis using the R package EdgeR. Graphical representations were generated utilizing the Bioinfokit and Matplotlib packages in Python.

## Data availability

Complete assemblies for Haloferax 48N sp. Atlit were deposited to NCBI genomes, accession numbers CP137689, CP137690, CP137691, CP137692, CP137693; this is a revision of the previous draft assembly, deposited in NCBI (PRJNA431124). Complete sequences of the virus LSV-48N and the plasmid pWL102 are available from GenBank, accession numbers PP151251 and PP297526, respectively. Raw sequence data from RNA-Seq, and from CRISPR spacer acquisition assays, are found in NCBIs SRA (Sequence Read Archive), project accession numbers PRJNA1072013 and PRJNA1072009, respectively. Custom code and scripts generated or used in this work are fully and freely available under the MIT open source license via https://github.com/UriNeri/HLSV1.

The source data of this paper are collected in the following database record: biostudies:S-SCDT-10_1038-S44319-025-00540-3.

## Peer review information

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

## Acknowledgements

The authors thank Shachar Robinzon for performing Haloferax mating experiments, Vered Holdengreber for help with electron microscopy and R. Thane Papke for his help with 48N genomics. This work was funded by a European Research Council (grant AdG 787514), and the Israeli Science Foundation (grant 1599/24).

## Author contributions

**Israela Turgeman-Grott**: Conceptualization; Supervision; Investigation; Methodology; Writing—original draft; Writing—review and editing. **Noam Golan**: Investigation; Methodology; Writing—review and editing. **Uri Neri**: Data curation; Investigation; Writing—original draft. **Doron Naki**: Investigation; Methodology. **Neta Altman-Price**: Methodology; Writing—review and editing. **Kim Eizenshtein**: Investigation. **Deepak K Choudhary**: Investigation. **Rachel Levy**: Investigation. **Sharon Navok**: Investigation; Methodology. **Lee Cohen**: Investigation. **Yarden Shalev**: Investigation; Methodology. **Himani Singla**: Investigation. **Leah Reshef**: Data curation; Software; Formal analysis; Investigation; Visualization; Writing—review and editing. **Uri Gophna**: Conceptualization; Data curation; Supervision; Funding acquisition; Investigation; Writing—original draft; Writing—review and editing.

Source data underlying figure panels in this paper may have individual authorship assigned. Where available, figure panel/source data authorship is listed in the following database record: biostudies:S-SCDT-10_1038-S44319-025-00540-3.

## Disclosure and competing interests statement

The authors have registered a patent covering the improvement of haloarchaeal strains by curing them of proviruses.

# Expanded View Figures

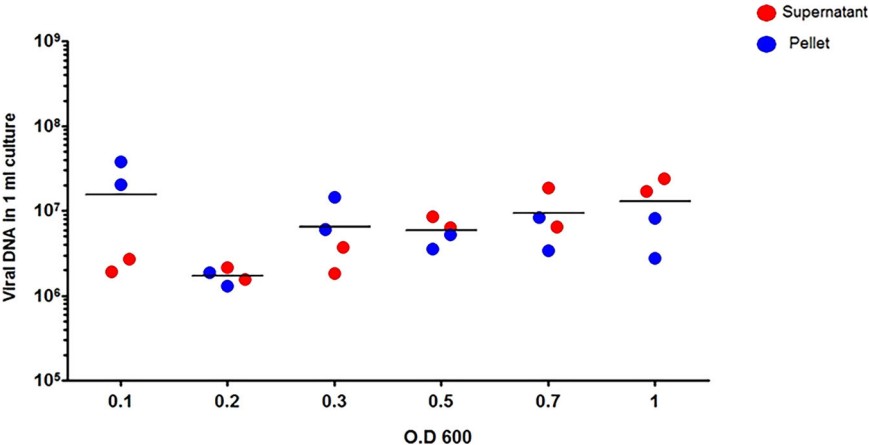

**Figure EV1. Digital droplet PCR quantification of LSV-48N genome copies along the growth curve (increasing OD).**

Taken from a culture of 48N grown on rich (YPC) medium. Two biological replicates are shown.

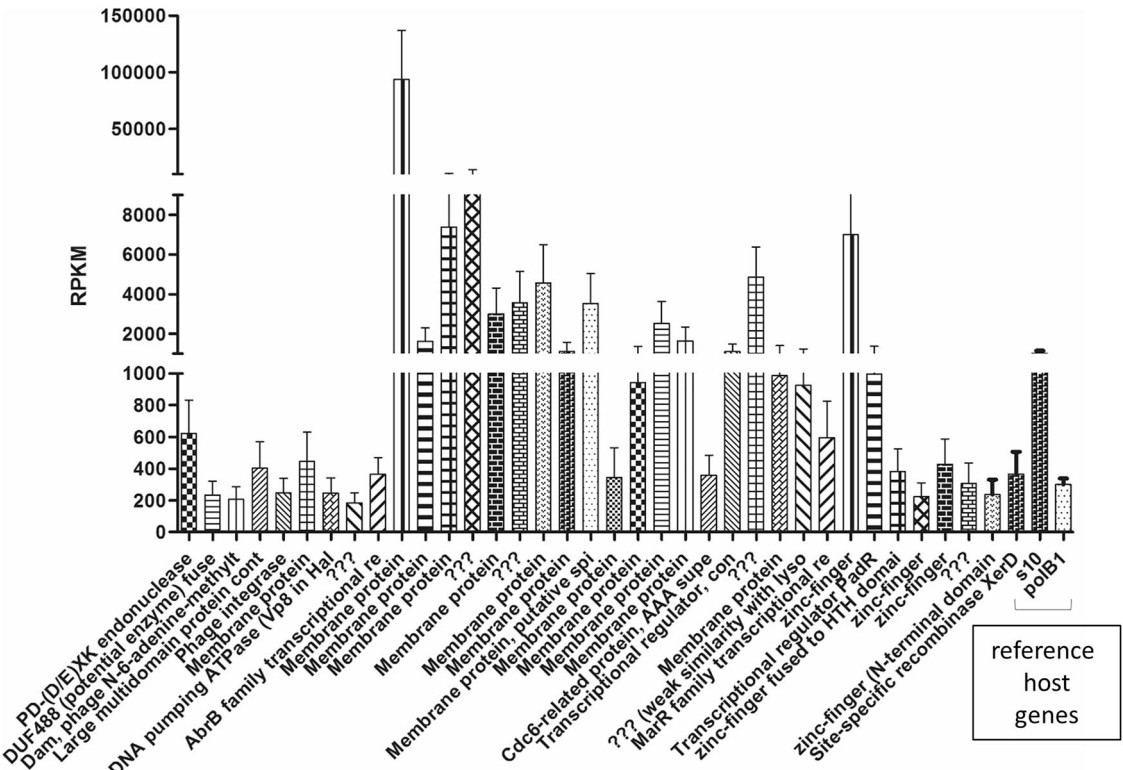

**Figure EV2. Mean gene expression levels of the LSV-48N genes in a virus-infected 48N culture.**

Based on three 3 RNA-Seq experiments (biological replicates). A highly-expressed host gene and a moderately-expressed host gene are provided as a reference. Genes of unknown function are marked as "???".

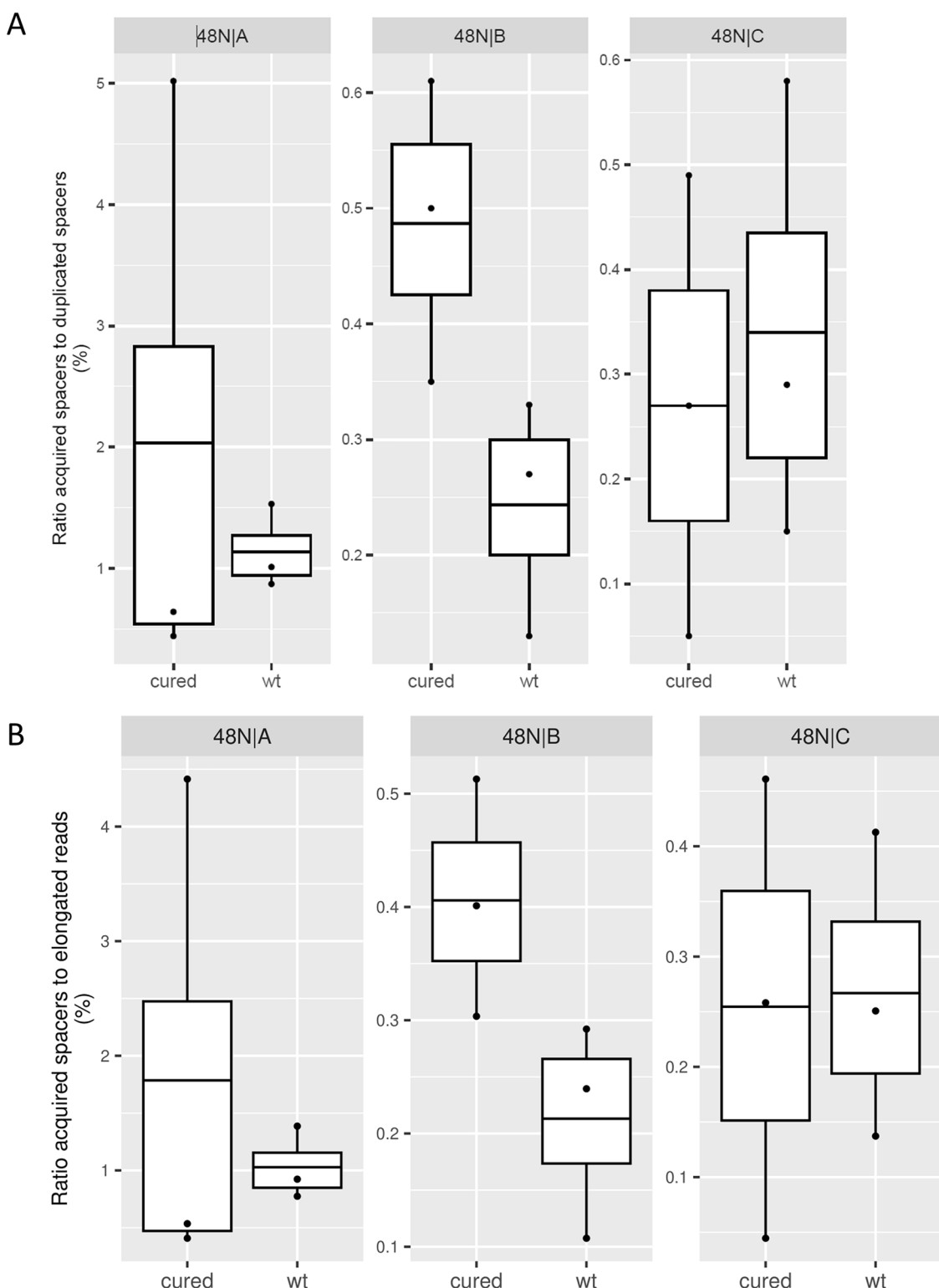

**Figure EV3. Spacer acquisition in the three CRISPR arrays, A, B and C, in 48N and its virus-cured derivative.**

Lines within bars are the mean of the three biological replicates, shown as dots. (A) The ratio of acquired spacers vs. duplicated spacers. (B) Ratio of acquired spacers vs. all reads of length corresponding to the length of an extended array, indicating a potential spacer acquisition event.

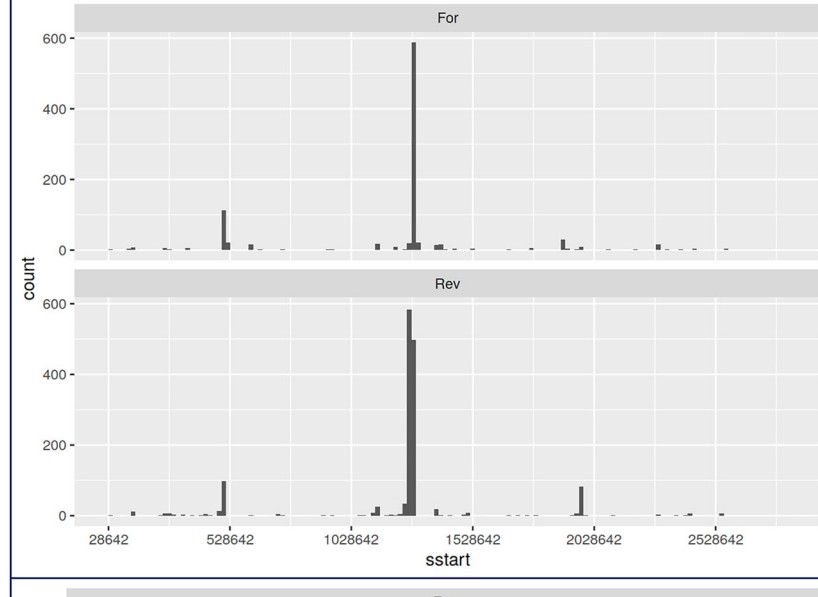

Chromosome

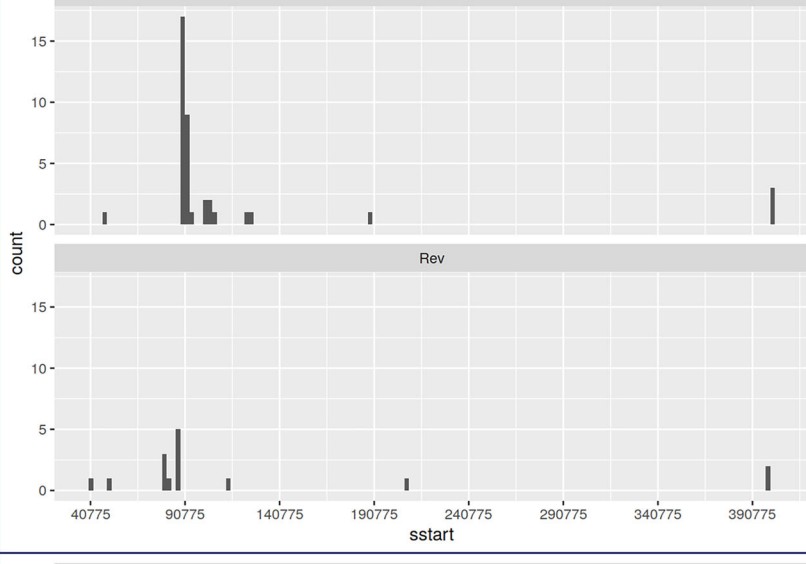

Plasmid 1

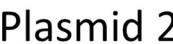

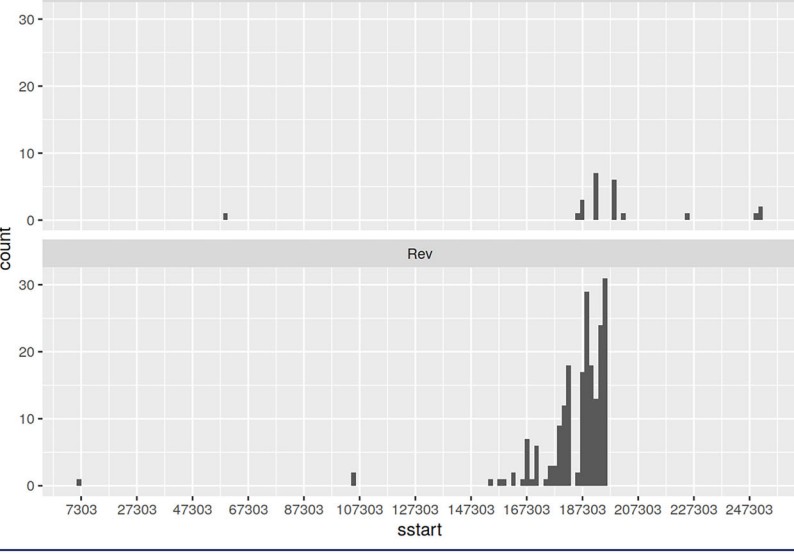

Plasmid 2

◀ **Figure EV4. Spacer acquisition event count by genomic location.**

Data include three biological replicates, over all three arrays in both strains.

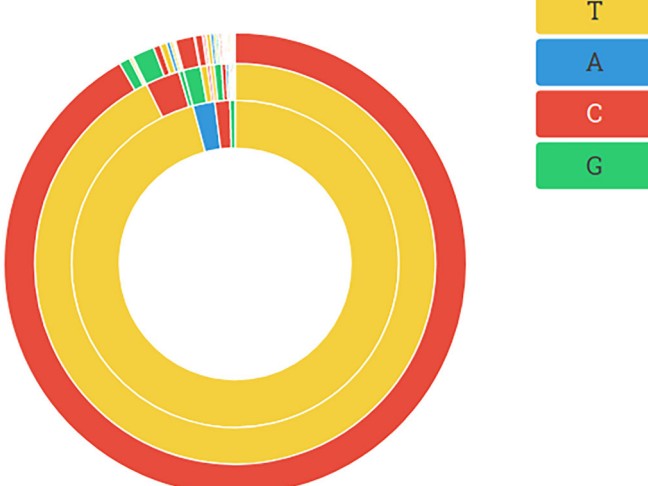

**Figure EV5. PAM wheel (Leenay et al, 2016) representation of protospacer adjacent motifs in the spacers acquired by 48N.**

Inner circle corresponds to the spacer-proximal nucleotide. Data include all new spacer acquisition events from the three biological replicates.

