## [Peer Review File · EMBO Reports]

A previously undescribed archaeal virus suppresses host immunity

Israella Turgeman-Grott, Noam Golan, Uri Neri, Doron Naki, Neta Altman-Price, Kim Eizenshtein, Deepak Choudhary, Rachel Levy, Sharon Navok, Yarden Shalev, Himani Himani, Leah Reshef, Uri Gophna, and Lee Cohen

Corresponding author(s): Uri Gophna (urigo@post.tau.ac.il)

Review Timeline:

Submission Date:	29th May 25
Editorial Decision:	5th Jun 25
Revision Received:	3rd Jul 25
Accepted:	18th Jul 25

Editor: Achim Breiling

Transaction Report: A revised version of this manuscript was transferred to EMBO reports following peer review at the EMBO Journal.

Dear Prof. Gophna,

Thank you for transferring your revised manuscript to EMBO reports. I now went again through the manuscript, and the referee reports from The EMBO Journal (attached again below). As you know, referee #1 was satisfied by the revision. However, referee #2 has remaining concerns, I ask you to address in a final revised manuscript.

Please tone down the conclusions and add further discussions to the manuscript text on the experimental challenges and alternative interpretations. Please also address in a final p-b-p-response and/or text changes the remaining concerns of referee #2, in particular the point regarding the potential alternative explanations for the observed changes in CRISPR spacer acquisition.

- Please provide individual production quality figure files as .eps, .tif, .jpg (one file per figure), of main figures and EV figures (without their legends). Please upload these as separate, individual files upon re-submission. Please make sure that all figure panels are called out separately and sequentially in the manuscript text

For more details please refer to our guide to authors:

See also our guide for figure preparation:

Moreover, please consult our guidelines for figure legend preparation:

- We updated our journal's competing interests policy in January 2022 and request authors to consider both actual and perceived competing interests. Please review the policy <https://www.embopress.org/competing-interests> and update your competing interests if necessary. Please name this section 'Disclosure and Competing Interests Statement' and put it after the Acknowledgements section.

- Please add up to 5 keywords to the manuscript and order the sections like this, using these names:

Title page - Abstract - Keywords - Introduction - Results & Discussion - Methods - Data availability section - Acknowledgements (please include here also the funding information) - Disclosure and Competing Interests Statement - References - Figure legends - Expanded View Figure legends

- Please provide a complete author checklist including the author information.

- Please check again that the number "n" for how many independent experiments were performed, their nature (biological versus technical replicates), the bars and error bars (e.g. SEM, SD) and the test used to calculate p-values is indicated in the respective figure legends (main, EV and Appendix figures). Please also check that all the p-values are explained in the legend, and that these fit to those shown in the figure. Please provide statistical testing where applicable. Please avoid the phrase 'independent experiment', but clearly state if these were biological or technical replicates. Please also indicate (e.g. with n.s.) if testing was performed, but the differences are not significant. In case n=2, please show the data as separate datapoints without error bars and statistics. See also:

<http://www.embopress.org/page/journal/14693178/authorguide#statisticalanalysis>

If n<5, please show single datapoints for diagrams. Moreover:

- Please note that information related to n is missing in the legends of figures 3C, D

- Please note that the error bars are not defined in the legend of figure 3C

- Please add scale bars of similar style and thickness to microscopic images, using clearly visible black or white bars (depending on the background). Please place these in the lower right corner of the images themselves. Please do not write on or near the bars in the image but define the size in the respective figure legend. Presently, error bars are often hard to see and have text

nearby. Please check.

- Please add specific URLs for the datasets CP137689-CP137693, PRJNA431124, PP151251, PP297526, PRJNA1072013, PRJNA1072009 to the data availability section.

- Per journal policy, we do not allow 'data not shown', which is stated in the manuscript (page 8). All data referred to in the paper should be displayed in the main or Expanded View figures, or an Appendix. Thus, please add these data (or change the text accordingly if these data are not central to the study). See:
<https://www.embopress.org/page/journal/14693178/authorguide#unpublisheddata>

- All Materials and Methods need to be described in the main text using our 'Structured Methods' format, which is required for all research articles. According to this format, the Methods section should include a Reagents and Tools Table (listing key reagents, experimental models, software, and relevant equipment and including their sources and relevant identifiers), uploaded as separate file, and a Methods section in which we encourage the authors to describe their methods using a step-by-step protocol format with bullet points, to facilitate the adoption of the methodologies across labs. More information on how to adhere to this format as well as downloadable templates (.doc) for the Reagents and Tools Table can be found in our author guidelines (section 'Structured Methods'):

- I would suggest to put the supplementary tables to the Appendix file (see above), using the name Appendix Table Sx. Large tables (e.g. T3, T4 and T6) are datasets. Please upload these as dataset files, named Dataset EVx (uploaded in excel format) and with their legends on the first TAB. Finally, please update all callouts.

- The movie files should be renamed 'Movie EV1' and the callout(s) updated. It needs a legend as readme text file ZIPed together with the movie file.

- Please remove all outdated files from the manuscript submission system.

In addition, I would need from you uploaded separately:

- a short, two-sentence summary of the manuscript (not more than 35 words).

- two to four short (!) bullet points highlighting the key findings of your study (two lines each).

- a schematic summary figure as separate file that provides a sketch of the major findings (not a data image) in jpeg or tiff format (with the exact width of 550 pixels and a height of not more than 400 pixels) that can be used as a visual synopsis on our website.

I look forward to seeing the final revised version of your manuscript when it is ready.

Please let me know if you have questions or comments regarding the revision.

Best,

Referee #1:

The authors have considerably improved the manuscript based on suggestions of the reviewers. They unfortunately discovered that their viral detection mutant was heterozygous concerning the pyrE gene. I think that it is very good that they honestly report on this. It also does not undermine their conclusion and they have adjusted the text accordingly. So the manuscript is acceptable for publication.

Referee #2:

The authors made their effort trying to improve the manuscript. Unfortunately the work still remains immature. The polyploidy nature makes the genetic purity of the cured-48N questionable. While it might be true that LSV was cured from the culture (very strange phenomenon, though), the cured "clone" may contain genetic variations in other genomic regions that contribute to the transcriptomic changes. The main conclusion is therefore not solidly supported. In addition, the statement in Abstract about

CRISPR spacer acquisition is not consistent with the analysis presented in Results (the main interference on CRISPR acquisition might be from another genetic element, not LSV).

Point-by-point

Please tone down the conclusions and add further discussions to the manuscript text on the experimental challenges and alternative interpretations.

We have added such an alternative explanation for the changes in gene expression (see our response to referee 2) to the conclusion, which tones it down quite a bit.

“Additionally, since virus curing resulted from an unclear molecular process (see Methods), it is possible that some changes to gene expression are the product of secondary heterozygous mutations that due to the highly polyploid nature of this organism, escaped detection by sequencing.”

We think the current version of the conclusions is very balanced, and does not overstate any of the findings. Similarly, the abstract has been revised to better reflect the results and the justified criticism of referee 2. “Nonetheless, even in the virus-cured background spacer acquisition is very low, indicating that another genetic element is disrupting CRISPR activity.”

Note that somehow this referee missed the section in the text comparing virus-cured and virus-infected spacer acquisition, in the result section, which ends with the following text:

“Thus, while LSV-48N somehow reduces the transcription of *cas* genes, it is probably another genetic element within the 48N genome that more strongly interferes with the spacer acquisition process of the 48N CRISPR-Cas system.”

Please also address in a final p-b-p-response and/or text changes the remaining concerns of referee #2, in particular the point regarding the potential alternative explanations for the observed changes in CRISPR spacer acquisition.

Done

- Please provide individual production quality figure files as .eps, .tif, .jpg (one file per figure), of main figures and EV figures (without their legends). Please upload these as separate, individual files upon re-submission. Please make sure that all figure panels are called out separately and sequentially in the manuscript text.

Done

The Expanded View format, which will be displayed in the main HTML of the paper in a collapsible format, has replaced the Supplementary information. You can submit up to 5 images as Expanded View. Please follow the nomenclature Figure EV1, Figure EV2 etc. The figure legend for these should be included in the main manuscript document file in a section called Expanded View Figure Legends after the main Figure Legends section.

Done

Additional Supplementary material should be supplied as a single pdf file labeled Appendix. The Appendix should have page numbers and needs to include a table of content on the first page (with page numbers) and legends for all content. Please follow the nomenclature Appendix Figure Sx, Appendix Table Sx etc. throughout the text, and also label the figures and tables according to this nomenclature.

For more details please refer to our guide to authors:

Done

See also our guide for figure preparation:

http://wol-prod-cdn.literatumonline.com/pb-assets/embosite/EMBOPress_Figure_Guidelines_061115-1561436025777.pdf

Moreover, please consult our guidelines for figure legend preparation:

Done

- We updated our journal's competing interests policy in January 2022 and request authors to consider both actual and perceived competing interests. Please review the policy <https://www.embopress.org/competing-interests> and update your competing interests if necessary. Please name this section 'Disclosure and Competing Interests Statement' and put it after the Acknowledgements section.

Done

- Please add up to 5 keywords to the manuscript and order the sections like this, using these names:

Title page - Abstract - Keywords - Introduction - Results & Discussion - Methods - Data availability section - Acknowledgements (please include here also the funding information) - Disclosure and Competing Interests Statement - References - Figure legends - Expanded View Figure legends

Done

- Please provide a complete author checklist including the author information.

Done

- Please check again that the number "n" for how many independent experiments were performed, their nature (biological versus technical replicates), the bars and error bars (e.g. SEM,

SD) and the test used to calculate p-values is indicated in the respective figure legends (main, EV and Appendix figures). Please also check that all the p-values are explained in the legend, and that these fit to those shown in the figure. Please provide statistical testing where applicable. Please avoid the phrase 'independent experiment', but clearly state if these were biological or technical replicates. Please also indicate (e.g. with n.s.) if testing was performed, but the differences are not significant. In case n=2, please show the data as separate datapoints without error bars and statistics. See also:

<http://www.embopress.org/page/journal/14693178/authorguide#statisticalanalysis>

If n<5, please show single datapoints for diagrams. Moreover:

- Please note that information related to n is missing in the legends of figures 3C, D

Fixed

- Please note that the error bars are not defined in the legend of figure 3C

I assume you meant 3B, fixed.

- Please add scale bars of similar style and thickness to microscopic images, using clearly visible black or white bars (depending on the background). Please place these in the lower right corner of the images themselves. Please do not write on or near the bars in the image but define the size in the respective figure legend. Presently, error bars are often hard to see and have text nearby. Please check.

Done

- Please add specific URLs for the datasets CP137689-CP137693, PRJNA431124, PP151251, PP297526, PRJNA1072013, PRJNA1072009 to the data availability section.

Done

- Per journal policy, we do not allow 'data not shown', which is stated in the manuscript (page 8).

Text was edited accordingly and now reads “Furthermore, it even grows faster than DS2 in the lab on standard media (Fig. 3B)”

- All Materials and Methods need to be described in the main text using our 'Structured Methods' format, which is required for all research articles. According to this format, the Methods section should include a Reagents and Tools Table (listing key reagents, experimental models, software, and relevant equipment and including their sources and relevant identifiers), uploaded as separate file, and a Methods section in which we encourage the authors to describe their methods using a step-by-step protocol format with bullet points, to facilitate the adoption of the methodologies across labs. More information on how to adhere to this format as well as downloadable templates (.doc) for the Reagents and Tools Table can be found in our author guidelines (section 'Structured Methods'):

Done

- I would suggest to put the supplementary tables to the Appendix file (see above), using the name Appendix Table Sx. Large tables (e.g. T3, T4 and T6) are datasets. Please upload these as dataset files, named Dataset EVx (uploaded in excel format) and with their legends on the first TAB. Finally, please update all callouts.

Done

- The movie files should be renamed 'Movie EV1' and the callout(s) updated. It needs a legend as readme text file ZIPed together with the movie file.

Done

- Please remove all outdated files from the manuscript submission system.

In addition, I would need from you uploaded separately:

I look forward to seeing the final revised version of your manuscript when it is ready.

Please let me know if you have questions or comments regarding the revision.

Best,

Referee #1:

The authors have considerably improved the manuscript based on suggestions of the reviewers. They unfortunately discovered that their viral detection mutant was heterozygous concerning the pyrE gene. I think that it is very good that they honestly report on this. It also does not undermine their conclusion and they have adjusted the text accordingly. So the manuscript is acceptable for publication.

Referee #2:

The authors made their effort trying to improve the manuscript. Unfortunately the work still remains immature. The polyploidy nature makes the genetic purity of the cured-48N questionable. While it might be true that LSV was cured from the culture (very strange phenomenon, though), the cured "clone" may contain genetic variations in other genomic regions that contribute to the transcriptomic changes. The main conclusion is therefore not solidly supported.

We have added the following caveat in the discussion:

“Additionally, since virus curing resulted from an unclear molecular process (see Methods), it is possible that some changes to gene expression are the product of secondary heterozygous mutations that due to the highly polyploid nature of this organism, escaped detection by sequencing.”

In addition, the statement in Abstract about CRISPR spacer acquisition is not consistent with the analysis presented in Results (the main interference on CRISPR acquisition might be from another genetic element, not LSV).

Abstract was revised and the relevant sentences now state:

“This virus subverts host defenses by reducing their transcription, including transcription of the CRISPR spacer acquisition machinery. Nonetheless, even in the virus-cured background spacer acquisition is very low, indicating that another genetic element is disrupting CRISPR activity.”

Prof. Uri Gophna
Tel Aviv University
Shmunis School of Biomedicine and Cancer Research
Haim Levanon 1
Tel Aviv 69978
Israel

Dear Prof. Gophna,

I am very pleased to accept your manuscript for publication in the next available issue of EMBO reports. Thank you for your contribution to our journal.

Yours sincerely,
